# Clinical Phenotypes of Dual Kidney Transplant Recipients in the United States as Identified through Machine Learning Consensus Clustering

**DOI:** 10.3390/medicina58121831

**Published:** 2022-12-12

**Authors:** Supawit Tangpanithandee, Charat Thongprayoon, Caroline C. Jadlowiec, Shennen A. Mao, Michael A. Mao, Pradeep Vaitla, Napat Leeaphorn, Wisit Kaewput, Pattharawin Pattharanitima, Pajaree Krisanapan, Pitchaphon Nissaisorakarn, Matthew Cooper, Wisit Cheungpasitporn

**Affiliations:** 1Division of Nephrology and Hypertension, Department of Medicine, Mayo Clinic, Rochester, MN 55905, USA; 2Chakri Naruebodindra Medical Institute, Faculty of Medicine Ramathibodi Hospital, Mahidol University, Samut Prakan 10540, Thailand; 3Division of Transplant Surgery, Mayo Clinic, Phoenix, AZ 85054, USA; 4Division of Transplant Surgery, Mayo Clinic, Jacksonville, FL 32224, USA; 5Division of Nephrology and Hypertension, Department of Medicine, Mayo Clinic, Jacksonville, FL 32224, USA; 6Division of Nephrology, University of Mississippi Medical Center, Jackson, MS 39216, USA; 7Department of Military and Community Medicine, Phramongkutklao College of Medicine, Bangkok 10400, Thailand; 8Division of Nephrology, Department of Internal Medicine, Thammasat University, Bangkok 12120, Thailand; 9Department of Medicine, Division of Nephrology, Massachusetts General Hospital, Harvard Medical School, Boston, MA 02114, USA; 10Medstar Georgetown Transplant Institute, Georgetown University School of Medicine, Washington, DC 21042, USA

**Keywords:** dual kidney transplant, dual kidney transplant recipients, transplant, transplantation, kidney transplantation, clustering, machine learning, artificial intelligence

## Abstract

*Background and Objectives*: Our study aimed to cluster dual kidney transplant recipients using an unsupervised machine learning approach to characterize donors and recipients better and to compare the survival outcomes across these various clusters. *Materials and Methods*: We performed consensus cluster analysis based on recipient-, donor-, and transplant-related characteristics in 2821 dual kidney transplant recipients from 2010 to 2019 in the OPTN/UNOS database. We determined the important characteristics of each assigned cluster and compared the post-transplant outcomes between clusters. *Results*: Two clinically distinct clusters were identified by consensus cluster analysis. Cluster 1 patients was characterized by younger patients (mean recipient age 49 ± 13 years) who received dual kidney transplant from pediatric (mean donor age 3 ± 8 years) non-expanded criteria deceased donor (100% non-ECD). In contrast, Cluster 2 patients were characterized by older patients (mean recipient age 63 ± 9 years) who received dual kidney transplant from adult (mean donor age 59 ± 11 years) donor with high kidney donor profile index (KDPI) score (59% had KDPI ≥ 85). Cluster 1 had higher patient survival (98.0% vs. 94.6% at 1 year, and 92.1% vs. 76.3% at 5 years), and lower acute rejection (4.2% vs. 6.1% within 1 year), when compared to cluster 2. Death-censored graft survival was comparable between two groups (93.5% vs. 94.9% at 1 year, and 89.2% vs. 84.8% at 5 years). *Conclusions*: In summary, DKT in the United States remains uncommon. Two clusters, based on specific recipient and donor characteristics, were identified through an unsupervised machine learning approach. Despite varying differences in donor and recipient age between the two clusters, death-censored graft survival was excellent and comparable. Broader utilization of DKT from high KDPI kidneys and pediatric en bloc kidneys should be encouraged to better address the ongoing organ shortage.

## 1. Introduction

Kidney transplantation is widely acknowledged as the most effective modality of renal replacement therapy for patients with end-stage kidney disease (ESKD). It not only increases expected patient survival as compared to chronic dialysis but also improves the quality of life [1,2,3]. However, there is a large gap between the number of patients receiving a transplant and the number of people waiting due to a limited number of donor organs [4,5]. Moreover, after the Organ Procurement and Transplantation Network (OPTN) introduced the kidney donor profile index (KDPI) as a new national deceased donor kidney allocation policy in 2013 [6], the kidney discard rate in the US greatly rose from 10% in 1998 to 21% in 2020 [7]. As a result, there have been ongoing efforts by the transplant community to increase the size of the deceased donor pool by using more kidneys from expanded criteria (ECD) and high KDPI donors, donation after cardiocirculatory death donors, as well as standard criteria donors with prolonged warm or cold ischemic times (CIT), acute kidney injury donors, and donors at the extremes of age [7]. Application of DKT of these discard at-risk groups has been another proposed strategy to further maximized transplant opportunities.

DKT is a strategy used in kidney transplantation to augment nephron mass in higher-risk transplant kidneys [8,9]. DKT is most often performed when donors are very young or very old and dual utilization allows compensation for healthy but limited renal mass or baseline chronic changes [10,11]. To date, reported graft survival outcomes in adult DKT compared to ECD single kidney transplantation (SKT) remain limited although several studies have been suggestive of outcomes with DKT [11,12,13,14,15,16,17,18,19,20,21,22]. Similarly, there continues to be heterogeneity for small pediatric donors with a dual versus single consideration recommended for those donors weighting >15 kg [11,23]. Although numerous studies have focused on defining optimal allocation criteria for pediatric and adult DKT, a machine learning technique has yet to be applied [12,17,18,19,20,24,25,26,27,28,29,30,31,32,33,34,35,36,37,38,39,40].

Machine learning is a subfield of artificial intelligence that has been applied to assist clinicians in making better clinical decisions in various areas of the medical field [41,42,43]. Generally, machine learning provides three types of algorithms: supervised, unsupervised, and reinforcement learning [41,42,43]. By identifying certain similarities and differences in various input variables, a computer system can complete a task without explicit programming. This is known as unsupervised machine learning [41,42,43,44]. As a result, grouping data into clinically relevant clusters can aid clinicians [42,43,45,46,47]. In this study, an unsupervised machine learning clustering approach was used to identify distinct clusters of DKT recipients and their clinical outcomes using the OPTN/UNOS database from 2010 through 2019.

## 2. Materials and Methods

### 2.1. Data Source and Study Population

We screened renal transplant patients the OPTN/UNOS database (2010 to 2019) to identify adult DKT patients in the United States. We excluded patients who received simultaneous kidney transplants with other organs. The study was approved by the Ethics Board of the Mayo Clinic (IRB ID: 21-007698).

### 2.2. Data Collection

We abstracted the comprehensive list of recipient-, donor-, and transplant-related characteristics, as shown in Table 1, to include in cluster analysis. KDPI was calculated based on donor age, height, weight, race, history of hypertension, diabetes mellitus, hepatitis C, cause of death, serum creatinine, and donor after cardiac death criteria. KDPI score ranges from 0–100% with higher score indicating lower quality donor kidneys. All variables had missing data of <5%, and multivariable imputation by chained equation (MICE) method was subsequently utilized [48].

### 2.3. Clustering Analysis

ML was utilized via an unsupervised consensus clustering analysis to categorize clinical phenotypes of DKT recipients [44]. To prevent producing an excessive number of clusters, we applied a subsampling parameter of 80% with 100 iterations and a number of potential clusters (k) ranging from 2 to 10. The optimal number of clusters was established by appraising the consensus matrix (CM) heat map, cumulative distribution function (CDF), and cluster-consensus plots with the within-cluster consensus scores. The average consensus value for all pairings of individual belonging to the same cluster was determined as the within-cluster consensus score, which ranged from 0 to 1 [49]. A closer value to 1 suggests more cluster stability. The full description of the consensus cluster algorithms utilized in this study are provided in Appendix A.

### 2.4. Outcomes

Post-transplant outcomes included (1) patient death, and (2) 1- and 5-year death-censored allograft loss, and (3) acute allograft rejection within 1 year post-transplant. Patients were censored for death at the last follow-up date reported to the OPTN/UNOS database, thus death-censored graft failure was defined as the need for dialysis or kidney retransplantation.

### 2.5. Statistical Analysis

After we categorized adult DKT patients by the consensus clustering algorithm, the characteristics and outcomes among the assigned clusters were compared. The difference in clinical characteristics was tested by utilizing Student’s *t*-test for continuous data and Chi-squared test for categorical data. The key characteristics of clusters were identified by applying the standardized mean difference between each cluster and the overall cohort with the pre-specified cut-off of >0.3. We demonstrated the risk of death-censored graft failure and patient death after kidney retransplant using Kaplan–Meier plot. We calculated the hazard ratio (HR) for death-censored graft loss, and mortality using Cox proportional hazard analysis. Because the date of rejection was not reported in the OPTN/UNOS database, we calculated the odds ratio (OR) for 1-year rejection using logistic regression analysis. We did not adjust the association of the assigned clusters with post-transplant outcomes for clinical characteristics because clinically distinct clusters were purposefully generated from the consensus clustering approach. R, version 4.0.3 (RStudio, Inc., Boston, MA, USA; http://www.rstudio.com/, accessed on 21 July 2021); ConsensusClusterPlus package (version 1.46.0) and the MICE command in R were used for consensus clustering analysis and for multivariable imputation by chained equation, respectively [48].

## 3. Results

From 2010 to 2019, there were total 158,367 kidney transplant recipients in the U.S. Of these, 2821 (1.8%) underwent DKT. Consensus clustering analysis was applied into these 2821 DKT recipients. The CDF plot demonstrated the consensus distributions for each cluster (Figure 1A). The delta area plot revealed the relative change in the area under the CDF curve (Figure 1B). The greatest changes in the area were identified between k = 2 and k = 4, after which point the relative rise in area decreased substantially. As shown in the CM heatmap (Figure 1C and Appendix A), the ML algorithm identified cluster 2 with distinct borders, demonstrating high cluster stability across repeated iterations. The mean cluster consensus score was highest in two clusters (Figure 2). Thus, consensus clustering analysis identified two clinically distinct clusters of DKT recipients.

### 3.1. Characteristics of Each DKT Cluster

There were 1875 (66%) patients in Cluster 1, and 946 (34%) patients in Cluster 2. Table 1 and Figure 3 demonstrated characteristics of DKT patients according to the assigned clusters. Cluster 1 patients was characterized by younger patients (mean recipient age 49 ± 13 years) who received DKT from pediatric (mean donor age 3 ± 8 years) non-expanded criteria deceased donor (100% non-ECD). In contrast, Cluster 2 patients was characterized by older patients (mean recipient age 63 ± 9 years) who received DKT from adult (mean donor age 59 ± 11 years) donor with a high KDPI score (59% had KDPI ≥ 85). In total, 70% and 30% of transplanted kidneys were from ECD and non-ECD deceased donor, respectively.

### 3.2. Post-Transplant Outcomes of Each DKT Clusters

Table 2 shows post-transplant outcomes according to the assigned clusters. Cluster 1 patients had higher patient survival (98.0% vs. 94.6% at 1 year, and 92.1% vs. 76.3% at 5 years) compared to cluster 2 patients (Figure 4A), and it was consistent across subgroup analysis (Table 3). Cluster 1 and cluster 2 had comparable death-censored graft survival (93.5% vs. 94.9% at 1 year, and 89.2% vs. 84.8% at 5 years) (Figure 4B). However, Cluster 2 might be associated with higher risk of death-censored graft failure in male kidney transplant recipients or donors (Table 3). Cluster 1 had less acute rejection (4.2% vs. 6.1% within 1 year) compared to cluster 2 patients.

## 4. Discussion

DKT, the transplantation of two kidneys from the same donor into a single recipient, has been utilized as an alternative approach to expand the available donor pool [7,11]. Many reports show that the graft survival rate was higher in the DKT recipients than in those single kidney transplant (SKT) with ECD or high KDPI kidney [11,12,15,19]. According to our study, DKT remains uncommon in the United States, accounting for only 1.8% of the overall kidney transplants. Recent OPTN allocation changes took effect in 2019 and were further affected by implementation of the 250 nautical mile fixed circle allocation [50]. Despite policy intent, there was a decrease in the number of dual kidney transplants performed albeit initial monitoring occurred during the COVID pandemic.

In this study, we use an unsupervised machine learning consensus clustering approach to categorize DKT into two different clusters based on recipient and donor characteristics in the OPTN/UNOS database. Cluster 1 patients, which accounted for nearly 70% of all DKT, were younger patients who received DKT from pediatric non-ECD donors. In contrast, cluster 2 patients were characterized by older patients who received DKT from adult donors with higher KDPI scores.

Cluster 2 patients received higher KDPI kidneys and had higher incidence of delayed graft function, and thus higher acute rejection was observed in cluster 2 recipients when compared to cluster 1 patients. While it is perhaps not unexpected that patient survival was better in cluster 1 given that patients in cluster 1 were significantly younger and less likely to be diabetic as compared to those in cluster 2, death-censored graft survival was; however, comparable between the two clusters. Early graft losses due to technical complications likely account for decreased graft survival in pediatric DKT recipients along with increased probability of seeing recurrent primary disease within the allograft [51]. By comparison, recipient characteristics in combination with lower expected longevity in higher KDPI kidneys likely accounts for similarities in patient survival and death-censored graft loss for cluster 2. The findings of this study illustrate excellent death-censored graft survival in both clusters. This reflects appropriate donor-recipient pairing and kidney utilization in the transplant community. These data align with what has been observed within the OPTN post-policy implementation where a higher proportion of dual kidney recipients were aged 65+ year whereas the proportion of pediatric en bloc kidney transplants for recipients aged 18–34 and 35–49 years notably increased [50].

There are some limitations in this study. Due to the registry nature of this cohort, there is lack of detail specific to exact causes for graft loss and death as well as specific detail related to donor-recipient pairing and center-specific criteria for DKT utilization. Additionally, lack of difference in death-censored graft loss between the 2 clusters can be explained by differences in recipient and donor characteristics, which are not necessarily novel. Although these clusters clinically different, and the application of machine learning is novel, there are limitations in how these data will enhance current clinical decision-making. Future studies applying supervised machine learning with prediction models based upon this initial data will be of greater utility in assessing predictors of survival for DKT.

To the best of our knowledge, this is the first machine learning approach specifically targeted at DKT. Two DKT clusters were identified using machine learning clustering methods without human intervention. The outcomes of our clustering approach, based on machine learning, support existing studies demonstrating the importance of donor-recipient pairing in DKT outcomes which also highlights opportunities to improve the kidney allocation system in the United States. There are likely existing opportunities to further expand DKT utilization within the transplant community, particularly for high KDPI kidneys. The application of ML consensus clustering approach in this study provides a novel understanding of unique phenotypes of DKT recipients in order to advance allocation systems to expand the donor pool. Given excellent outcomes among both clusters, DKT from high KDPI kidneys and pediatric en bloc kidneys should be encouraged to better address the ongoing organ shortage.

## 5. Conclusions

In summary, DKT in the United States remains uncommon. Two clusters, based on specific recipient and donor characteristics, were identified through an unsupervised machine learning approach. Despite varying differences in donor and recipient age between the two clusters, death-censored graft survival was excellent and comparable. Broader utilization of DKT from high KDPI kidneys and pediatric en bloc kidneys should be encouraged to better address the ongoing organ shortage.

## Figures and Tables

**Figure 1 medicina-58-01831-f001:**
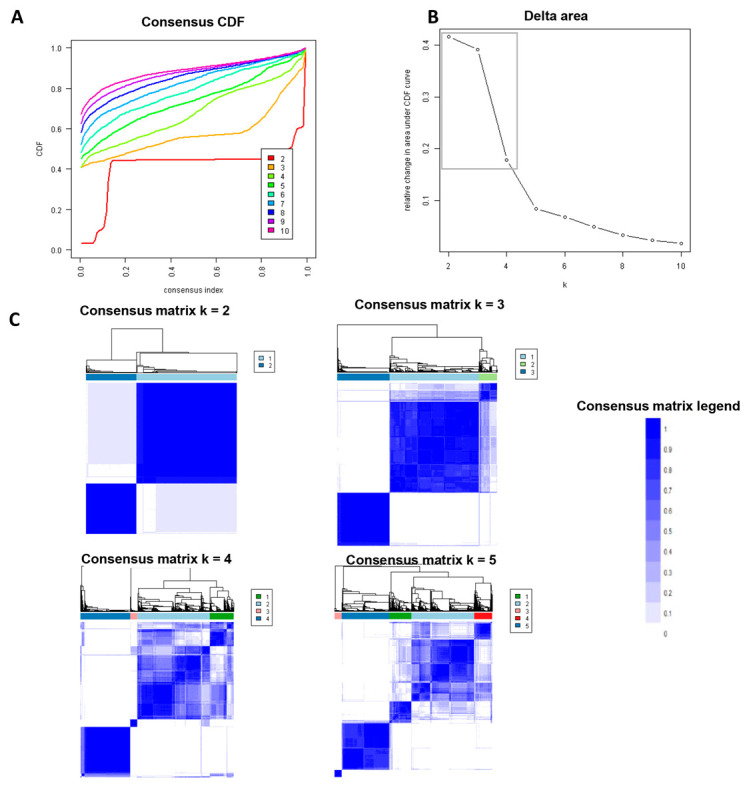
(**A**) Consensus CDF plot; (**B**) Delta area plot; (**C**) Consensus matrix heat map.

**Figure 2 medicina-58-01831-f002:**
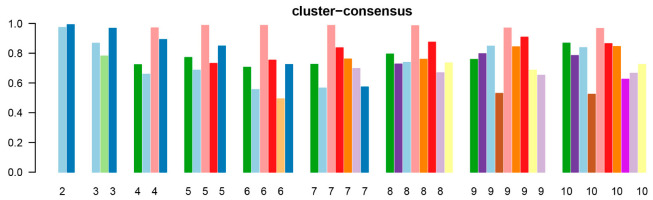
Mean consensus score for different numbers of clusters.

**Figure 3 medicina-58-01831-f003:**
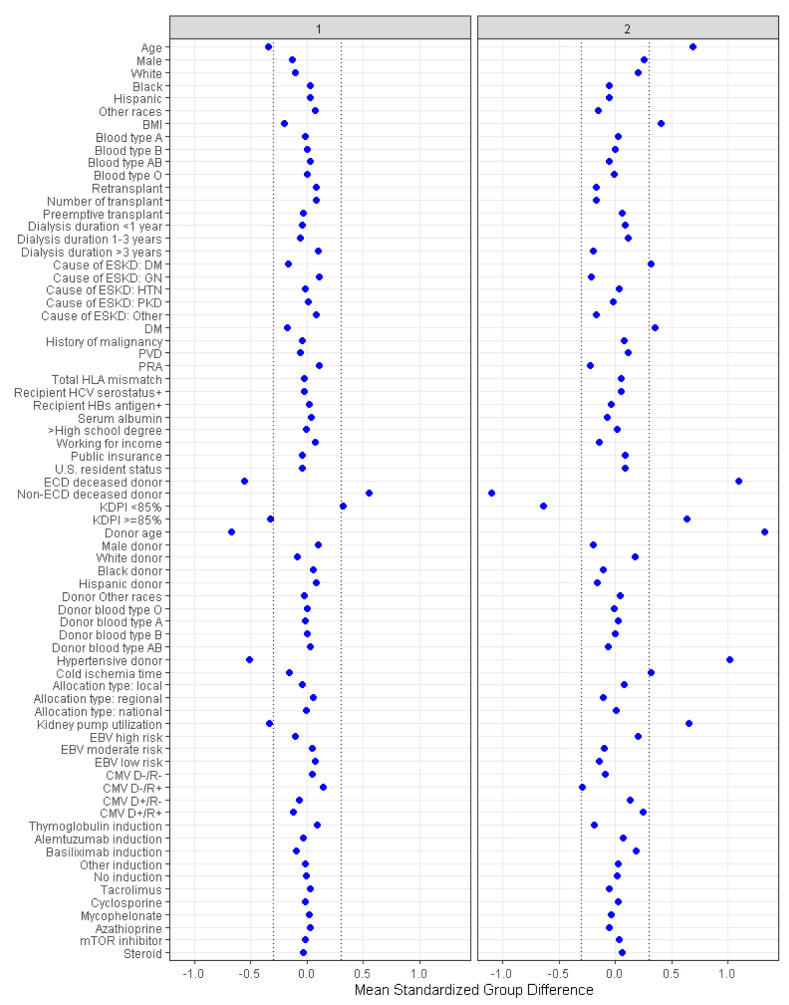
The standardized differences across two DKT clusters for each of baseline parameters.

**Figure 4 medicina-58-01831-f004:**
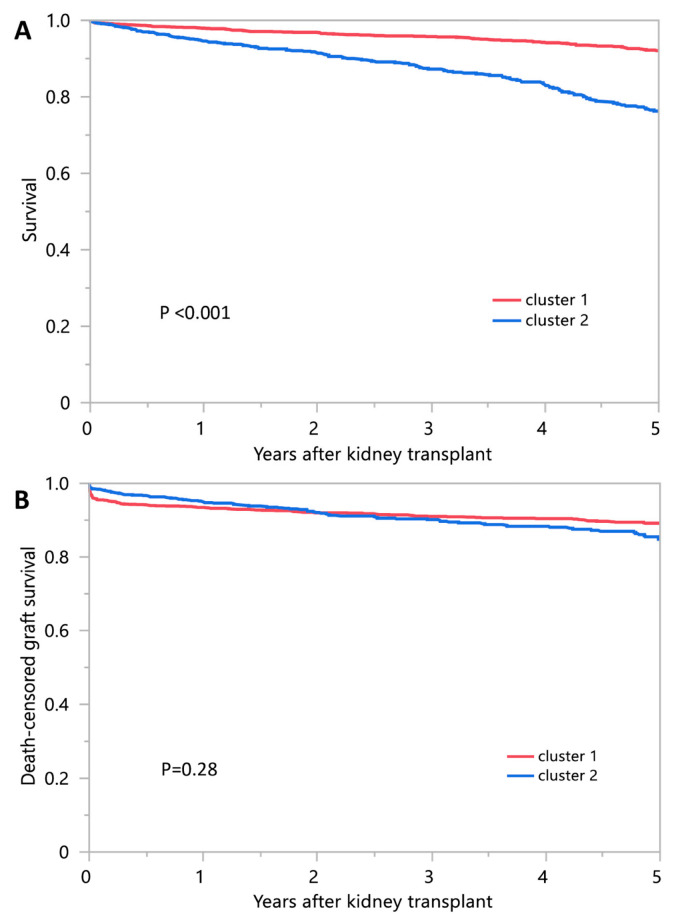
(**A**) Patient survival; (**B**) Death-censored graft survival outcomes among two identified clusters of DKT patients.

**Table 1 medicina-58-01831-t001:** Clinical characteristics of DKT patients according to the assigned clusters.

	All (n = 2821)	Cluster 1 (n = 1875)	Cluster 2 (n = 946)	*p*-Value
Recipient Age (year)	53.7 ± 13.8	48.9 ± 13.4	63.2 ± 8.8	<0.001
Recipient male sex	1511 (54)	882 (47)	629 (66)	<0.001
Recipient race				<0.001
- White	1038 (37)	596 (32)	442 (47)	
- Black	815 (29)	565 (30)	250 (26)	
- Hispanic	494 (18)	346 (18)	148 (16)	
- Other	474 (17)	368 (20)	106 (11)	
ABO blood group				0.239
- A	955 (34)	621 (33)	334 (35)	
- B	431 (15)	287 (15)	144 (15)	
- AB	144 (5)	106 (6)	38 (4)	
- O	1291 (46)	861 (46)	430 (45)	
Body mass index (kg/m^2^)	26.2 ± 4.7	25.3 ± 4.4	28.2 ± 4.8	<0.001
Kidney retransplant	131 (4.6)	120 (6)	11 (1)	<0.001
Dialysis duration				<0.001
- Preemptive	297 (11)	180 (10)	117 (12)	
- <1 year	308 (11)	179 (10)	129 (14)	
- 1–3 years	809 (29)	489 (26)	320 (34)	
- >3 years	1407 (50)	1027 (55)	380 (40)	
Cause of ESKD				<0.001
- Diabetes mellitus	746 (26)	362 (19)	384 (41)	
- Hypertension	738 (26)	477 (25)	261 (28)	
- Glomerular disease	626 (22)	499 (27)	127 (13)	
- Polycystic kidney disease	232 (8)	159 (8)	73 (8)	
- Other	479 (17)	378 (20)	101 (11)	
Comorbidity				
- Diabetes mellitus	920 (33)	455 (24)	465 (49)	<0.001
- Malignancy	233 (8)	135 (7)	98 (10)	0.004
- Peripheral vascular disease	214 (8)	113 (6)	101 (11)	<0.001
Panel reactive antibody, median (IQR)	0 (0, 11)	0 (0, 23)	0 (0, 0)	<0.001
Positive Hepatitis C virus serostatus	97 (3)	55 (3)	42 (4)	0.038
Positive Hepatitis B surface antigen	64 (2)	48 (3)	16 (2)	0.144
Positive Human immunodeficiency virus serostatus	28 (1)	22 (1)	6 (1)	0.173
Functional status				0.267
- 10–30%	7 (0)	5 (0)	2 (0.21)	
- 40–70%	1116 (40)	722 (39)	394 (42)	
- 80–100%	1698 (60)	1148 (61)	550 (58)	
Working income	771 (27)	571 (30)	200 (21)	<0.001
Public insurance	2125 (75)	1376 (73)	749 (79)	0.001
US resident	2776 (98)	1835 (97)	941 (99)	0.001
Undergraduate education or above	1434 (51)	946 (50)	488 (52)	0.570
Serum albumin (g/dL)	4 ± 0.6	4.0 ± 0.6	3.9 ± 0.6	0.005
Kidney donor status				<0.001
- Non-ECD deceased	2152 (76)	1872 (100)	280 (30)	
- ECD deceased	669 (24)	3 (0)	666 (70)	
Donor age	21.9 ± 28.1	3.0 ± 8.0	59.3 ± 10.9	<0.001
Donor male sex	1482 (53)	1079 (58)	403 (43)	<0.001
Donor race				<0.001
- White	1724 (61)	1063 (57)	661 (70)	
- Black	555 (20)	410 (22)	145 (15)	
- Hispanic	386 (14)	308 (16)	78 (8)	
- Other	156 (6)	94 (5)	62 (7)	
History of hypertension in donor	651 (23)	27 (1)	624 (66)	<0.001
Kidney donor profile index (KDPI)				<0.001
- KDPI < 85	1981 (70)	1593 (85)	388 (41)	
- KDPI ≥ 85	840 (30)	282 (15)	558 (59)	
HLA mismatch, median (IQR)	5 (4, 5)	5 (4, 5)	5 (4, 5)	0.06
Cold ischemia time (hours)	19.6 ± 10.1	18.0 ± 9.3	22.9 ± 10.9	<0.001
Kidney on pump	1028 (36)	385 (21)	643 (68)	<0.001
Delay graft function	660 (23)	324 (17)	336 (36)	<0.001
Allocation type				0.001
- Local	1509 (53)	965 (51)	544 (58)	
- Regional	654 (23)	476 (25)	178 (19)	
- National	658 (23)	434 (23)	224 (24)	
Epstein–Barr virus status				<0.001
- Low risk	64 (2)	63 (3)	1 (0)	
- Moderate risk	2588 (92)	1746 (93)	842 (89)	
- High risk	169 (6)	66 (4)	103 (11)	
Cytomegalovirus status				<0.001
- D−/R−	380 (13)	280 (15)	100 (11)	
- D−/R+	954 (34)	763 (41)	191 (20)	
- D+/R+	1084 (38)	608 (32)	476 (50)	
- D+/R−	403 (14)	224 (12)	179 (19)	
Induction immunosuppression				
- Thymoglobulin	1732 (61)	1236 (66)	496 (52)	<0.001
- Alemtuzumab	353 (13)	213 (11)	140 (15)	0.009
- Basiliximab	582 (21)	315 (17)	267 (28)	<0.001
- Other	87 (3)	54 (3)	33 (3)	0.378
- No induction	173 (6)	111 (6)	62 (7)	0.508
Maintenance Immunosuppression				
- Tacrolimus	2557 (91)	1713 (91)	844 (89)	0.065
- Cyclosporine	29 (1)	17 (1)	12 (1)	0.368
- Mycophenolate	2604 (92)	1739 (93)	865 (91)	0.218
- Azathioprine	7 (0)	7 (0)	0 (0)	0.060
- mTOR inhibitors	9 (0)	4 (0)	5 (1)	0.161
- Steroid	1954 (69)	1270 (68)	684 (72)	0.013

Abbreviations: D: Donor, ECD: Extended criteria donor, ESKD: end stage kidney disease, mTOR: Mammalian target of rapamycin, R: Recipient.

**Table 2 medicina-58-01831-t002:** Clinical outcomes.

	Cluster 1	Cluster 2	*p*-Value
1-year survival	98.0%	94.6%	<0.001
HR for 1-year mortality	1 (ref)	2.62 (1.70–4.08)	<0.001
5-year survival	92.1%	76.3%	<0.001
HR for 5-year mortality	1 (ref)	3.12 (2.41–4.05)	<0.001
1-year death-censored graft survival	93.5%	94.9%	0.08
HR for 1-year death-censored graft loss	1 (ref)	0.73 (0.51–1.03)	0.08
5-year death-censored graft survival	89.2%	84.8%	0.28
HR for 5-year death-censored graft loss	1 (ref)	1.15 (0.89–1.48)	0.28
1-year acute rejection	4.2%	6.1%	0.03
OR for 1-year acute rejection	1 (ref)	1.48 (1.05–2.10)	0.03

**Table 3 medicina-58-01831-t003:** Subgroup analysis.

	Patient Death		Death-Censored Graft Failure	
	HR (95% CI)	*p*-Value	HR (95% CI)	*p*-Value
Recipient age				
- <60	1.94 (1.13–3.21)	0.02	1.26 (0.85–1.83)	0.25
- ≥60	2.00 (1.42–2.87)	<0.001	1.41 (0.91–2.21)	0.12
Recipient sex				
- Male	2.92 (2.11–4.10)	<0.001	1.47 (1.03–2.10)	0.03
- Female	3.03 (1.95–4.68)	<0.001	1.00 (0.66–1.46)	0.99
Recipient race				
- White	3.12 (2.11–4.69)	<0.001	1.03 (0.68–1.55)	0.89
- Non-white	2.89 (2.04–4.12)	<0.001	1.25 (0.90–1.72)	0.18
Recipient body mass index				
- <30	3.20 (2.36–4.34)	<0.001	1.11 (0.81–1.50)	0.52
- ≥30	2.22 (1.34–3.82)	0.002	1.05 (0.64–1.74)	0.84
Kidney retransplant				
- No	3.25 (2.49–4.27)	<0.001	1.14 (0.88–1.48)	0.33
- Yes	3.59 (0.81–11.55)	0.09	2.95 (0.68–9.18)	0.13
Preemptive transplant				
- No	3.08 (2.35–4.03)	<0.001	1.21 (0.92–1.57)	0.16
- Yes	4.43 (1.64–13.94)	0.003	0.71 (0.25–1.78)	0.48
Recipient diabetes				
- No	3.61 (2.50–5.22)	<0.001	1.15 (0.82–1.60)	0.41
- Yes	1.84 (1.27–2.69)	0.001	1.11 (0.73–1.70)	0.63
PRA				
- 0	3.15 (2.31–4.33)	<0.001	1.14 (0.84–1.54)	0.39
- >0	3.13 (1.94–5.05)	<0.001	1.22 (0.74–1.94)	0.42
Donor sex				
- Male	3.46 (2.46–4.89)	<0.001	1.49 (1.05–2.08)	0.02
- Female	2.99 (2.01–4.50)	<0.001	0.90 (0.61–1.33)	0.60
Donor race				
- White	3.12 (2.23–4.38)	<0.001	1.02 (0.74–1.40)	0.91
- Non-White	3.23 (2.12–4.91)	<0.001	1.44 (0.95–2.19)	0.10
Donor hypertension				
- No	2.92 (2.03–4.13)	<0.001	1.11 (0.75–1.61)	0.58
- Yes	2.52 (0.80–15.30)	0.13	0.77 (0.32–2.52)	0.61
KDPI				
- <85	2.30 (1.56–3.33)	<0.001	1.21 (0.82–1.73)	0.33
- ≥85	3.12 (1.94–5.34)	<0.001	0.69 (0.46–1.03)	0.07

## Data Availability

Data will be made available by the authors upon reasonable request.

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
