# Peer review of "Clinical Phenotypes of Dual Kidney Transplant Recipients in the United States as Identified through Machine Learning Consensus Clustering"

_medicina, 2022, doi:10.3390/medicina58121831_

Round 1

Reviewer 1 Report

 Supawit Tangpanithandee and Colleagues present the results of unsupervised machine learning analysis of double kidney transplants in a registry study from US data. The results are interesting, individuating two groups, with broad differences in terms of both recipients and donor characteristics. although is it of interest that the two groups do not differ for death censored graft survival, more explanations and analyses should be given. i.e. graft and patients survival differences inside each group, as for example dividing according to KDPI, or comorbidities as cardiovascular diseases and/or diabetes of the donors and recipients, these kind of analyses can be given also without the need of machine learning, but can add important informations and help in the soundness of the paper. 

moreover, authors do not put so much emphasis on the novelty on the use of machine learning and the possible applications and benefits in the daily allocation systems to expand the donor pool. 

I think more analyses should be provided to publish the paper.

Author Response

Response to Reviewer#1

Comment #1

The results are interesting, individuating two groups, with broad differences in terms of both recipients and donor characteristics. although is it of interest that the two groups do not differ for death censored graft survival, more explanations and analyses should be given. i.e. graft and patients survival differences inside each group, as for example dividing according to KDPI, or comorbidities as cardiovascular diseases and/or diabetes of the donors and recipients, these kind of analyses can be given also without the need of machine learning, but can add important informations and help in the soundness of the paper. 

Response: Thankyou for reviewing our manuscripts and critical evaluation. We compared patient survival and death-censored graft survival in various subgroups as suggested, and added the result of subgroup analysis in Table 3.

The following statements in bold have been added to the result section.

Cluster 1 patients had higher patient survival (98.0% vs. 94.6% at 1 year, and 92.1% vs. 76.3% at 5 years) compared to Cluster 2 patients (Figure 3A), and it was consistent across subgroup analysis (Table 3). Cluster 1 and Cluster 2 had comparable death-censored graft survival (93.5% vs. 94.9% at 1 year, and 89.2% vs. 84.8% at 5 years) (Figure 3B). However, Cluster 2 might be associated with higher risk of death-censored graft failure in male kidney transplant recipients or donors (Table 3).

Table 3 Subgroup analysis

Patient death

Death-censored graft failure

HR (95% CI)

p-value

HR (95% CI)

p-value

Recipient age

-           <60

1.94 (1.13-3.21)

0.02

1.26 (0.85-1.83)

0.25

-           ≥60

2.00 (1.42-2.87)

<0.001

1.41 (0.91-2.21)

0.12

Recipient sex

-           Male

2.92 (2.11-4.10)

<0.001

1.47 (1.03-2.10)

0.03

-           Female

3.03 (1.95-4.68)

<0.001

1.00 (0.66-1.46)

0.99

Recipient race

-           White

3.12 (2.11-4.69)

<0.001

1.03 (0.68-1.55)

0.89

-           Non-white

2.89 (2.04-4.12)

<0.001

1.25 (0.90-1.72)

0.18

Recipient body mass index

-           <30

3.20 (2.36-4.34)

<0.001

1.11 (0.81-1.50)

0.52

-           ≥30

2.22 (1.34-3.82)

0.002

1.05 (0.64-1.74)

0.84

Kidney retransplant

-           No

3.25 (2.49-4.27)

<0.001

1.14 (0..88-1.48)

0.33

-           Yes

3.59 (0.81-11.55)

0.09

2.95 (0.68-9.18)

0.13

Preemptive transplant

-           No

3.08 (2.35-4.03)

<0.001

1.21 (0.92-1.57)

0.16

-           Yes

4.43 (1.64-13.94)

0.003

0.71 (0.25-1.78)

0.48

Recipient diabetes

-           No

3.61 (2.50-5.22)

<0.001

1.15 (0.82-1.60)

0.41

-           Yes

1.84 (1.27-2.69)

0.001

1.11 (0.73-1.70)

0.63

PRA

-           0

3.15 (2.31-4.33)

<0.001

1.14 (0.84-1.54)

0.39

-           >0

3.13 (1.94-5.05)

<0.001

1.22 (0.74-1.94)

0.42

Donor sex

-           Male

3.46 (2.46-4.89)

<0.001

1.49 (1.05-2.08)

0.02

-           Female

2.99 (2.01-4.50)

<0.001

0.90 (0.61-1.33)

0.60

Donor race

-           White

3.12 (2.23-4.38)

<0.001

1.02 (0.74-1.40)

0.91

-           Non-White

3.23 (2.12-4.91)

<0.001

1.44 (0.95-2.19)

0.10

Donor hypertension

-           No

2.92 (2.03-4.13)

<0.001

1.11 (0.75-1.61)

0.58

-           Yes

2.52 (0.80-15.30)

0.13

0.77 (0.32-2.52)

0.61

KDPI

-           <85

2.30 (1.56-3.33)

<0.001

1.21-0.82-1.73)

0.33

-           ≥85

3.12 (1.94-5.34)

<0.001

0.69 (0.46-1.03)

0.07

Comment #2

Authors do not put so much emphasis on the novelty on the use of machine learning and the possible applications and benefits in the daily allocation systems to expand the donor pool. 

Response: We appreciate the reviewer’s important comments and we agree with the reviewer. Thus, we additionally emphasize the novelty to help expand the donor pool as suggested. The following text has been added as suggested.

“The application of ML consensus clustering approach in this study provides a novel un-derstanding of unique phenotypes of DKT recipients in order to advance allocation sys-tems to expand the donor pool. Given excellent outcomes among both clusters, DKT from high KDPI kidneys and pediatric en bloc kidneys should be encouraged to better address the ongoing organ shortage.”

Thank you for your time and consideration.  We greatly appreciated the reviewer's and editor's time and comments to improve our manuscript. The manuscript has been improved considerably by the suggested revisions.

Reviewer 2 Report

The authors conducted an intersting study investigating whether transplant recipient baseline characteristics and KDPI effect graft survival in dual kidney transplant recipients using machine learning clustering. The manuscript is well written and the results well presented. I have only some minor comments which may improve the quality of the manuscript:

1. Please add the KDPI abbreviation in the Abstract.

2.  Please add how KDPI score was defined in the Methodology section.

3. Please add the p values in Table 2.

4.  How the authors explain the statistical difference regarding acute rejection at 1 year between the two groups?

5. Why the authors did not include the 5-year acute rejection in their outcomes?

Author Response

Response to Reviewer#2

The authors conducted an interesting study investigating whether transplant recipient baseline characteristics and KDPI effect graft survival in dual kidney transplant recipients using machine learning clustering. The manuscript is well written and the results well presented. I have only some minor comments which may improve the quality of the manuscript:

Response: Thankyou for reviewing our manuscripts and critical evaluation.

Comment #1

Please add the KDPI abbreviation in the Abstract.

Response: We added the abbreviation of KDPI in the abstract.

Comment #2

Please add how KDPI score was defined in the Methodology section.

Response: We agree with the reviewer. The following statements have been added to the method section.

“KDPI was calculated based on donor age, height, weight, race, history of hypertension, diabetes mellitus, hepatitis C, cause of death, serum creatinine, and donor after cardiac death criteria. KDPI score ranges from 0-100% with higher score indicating lower quality donor kidneys.”

Comment #3

Please add the p values in Table 2.

Response: We agree and we added p-value in Table 2 as suggested.

Comment #4

How the authors explain the statistical difference regarding acute rejection at 1 year between the two groups?

Response: The reviewer raises important point. We agree and thus added the discussion on the higher acute rejection among cluster 2 as suggested. The following text has been added as suggested.

“Cluster 2 patients received higher KDPI kidneys and had higher incidence of delayed graft function, and thus higher acute rejection was observed in cluster 2 recipients when compared to cluster 1 patients.”

Comment #5

Why the authors did not include the 5-year acute rejection in their outcomes?

Response: The reviewer raises the important point. We could not include 5-year acute rejection in one of the post-transplant outcomes because UNOS database only reported 1-year acute rejection but did not report 5-year rejection.

Thank you for your time and consideration.  We greatly appreciated the reviewer's and editor's time and comments to improve our manuscript. The manuscript has been improved considerably by the suggested revisions.
